# Directed-Info GAIL: Learning Hierarchical Policies from Unsegmented Demonstrations using Directed Information

**Mohit Sharma**\*, **Arjun Sharma**\*, **Nick Rhinehart, Kris M. Kitani**
Robotics Institute
Carnegie Mellon University
Pittsburgh, PA 15213, USA
`{mohits1,arjuns2,nrhineha,kkitani}@cs.cmu.edu`

## Abstract

The use of imitation learning to learn a single policy for a complex task that has multiple modes or hierarchical structure can be challenging. In fact, previous work has shown that when the modes are known, learning separate policies for each mode or sub-task can greatly improve the performance of imitation learning. In this work, we discover the interaction between sub-tasks from their resulting state-action trajectory sequences using a directed graphical model. We propose a new algorithm based on the generative adversarial imitation learning framework which automatically learns sub-task policies from unsegmented demonstrations. Our approach maximizes the directed information flow in the graphical model between sub-task latent variables and their generated trajectories. We also show how our approach connects with the existing Options framework, which is commonly used to learn hierarchical policies.

## 1 Introduction

Complex human activities can often be broken down into various simpler sub-activities or sub-tasks that can serve as the basic building blocks for completing a variety of complicated tasks. For instance, when driving a car, a driver may perform several simpler sub-tasks such as driving straight in a lane, changing lanes, executing a turn and braking, in different orders and for varying times depending on the source, destination, traffic conditions *etc.* Using imitation learning to learn a single monolithic policy to represent a structured activity can be challenging as it does not make explicit the sub-structure between the parts within the activity. In this work, we develop an imitation learning framework that can learn a policy for each of these sub-tasks given unsegmented activity demonstrations and also learn a macro-policy which dictates switching from one sub-task policy to another. Learning sub-task specific policies has the benefit of shared learning. Each such sub-task policy also needs to specialize over a restricted state space, thus making the learning problem easier.

Previous works in imitation learning (Li et al., 2017; Hausman et al., 2017) focus on learning each sub-task specific policy using *segmented* expert demonstrations by modeling the variability in each sub-task policy using a latent variable. This latent variable is inferred by enforcing high mutual information between the latent variable and expert demonstrations. This information theoretic perspective is equivalent to the graphical model shown in Figure 1 (Left), where the node $c$ represents the latent variable. However, since learning sub-task policies requires isolated demonstrations for each sub-task, this setup is difficult to scale to many real world scenarios where providing such segmented trajectories is cumbersome. Further, this setup does not learn a macro-policy to combine the learned sub-task policies in meaningful ways to achieve different tasks.

In our work, we aim to learn each sub-task policy directly from *unsegmented* activity demonstrations. For example, given a task consisting of three sub-tasks — A, B and C, we wish to learn a policy to complete sub-task A, learn when to transition from A to B, finish sub-task B and so on. To achieve this we use a causal graphical model, which can be represented as a Dynamic Bayesian Network as

---

\*Denotes equal contribution

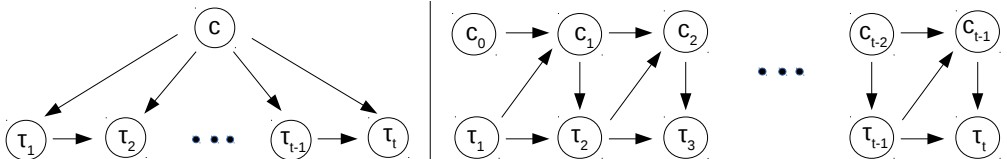

Figure 1: **Left:** Graphical model used in Info-GAIL Li et al. (2017). **Right:** Causal model in this work. The latent code causes the policy to produce a trajectory. The current trajectory, and latent code produce the next latent code

shown in Figure 1 (Right). The nodes $c_t$ denote latent variables which indicate the currently active sub-task and the nodes $\tau_t$ denote the state-action pair at time $t$. We consider as given, a set of expert demonstrations, each of which is represented by $\boldsymbol{\tau} = \{\tau_1, \cdots, \tau_T\}$ and has a corresponding sequence of latent factors $\boldsymbol{c} = \{c_1, \cdots, c_{T-1}\}$. The sub-activity at time $t$ dictates what state-action pair was generated at time $t$. The previous sub-task and the current state together cause the selection of the next sub-task.

As we will discuss in Section 3, extending the use of mutual information to learn sub-task policies from unsegmented demonstrations is *problematic*, as it requires learning the macro-policy as a conditional probability distribution which depends on the *unobserved future*. This unobserved future is unknown during earlier points of interaction (Figure 1). To alleviate this, in our work we aim to force the policy to generate trajectories that maximize the directed information or causal information (Massey, 1990) flow from trajectories to latent factors of variation within the trajectories instead of mutual information. Using directed information requires us to learn a causally conditioned probability distribution (Kramer, 1998) which depends only on the *observed past* while allowing the unobserved future to be sequentially revealed. Further, since there exists *feedback* in our causal graphical model *i.e.*, information flows from the latent variables to trajectories and vice versa, directed information also provides a better upper bound on this information flow between the latent variables and expert trajectories than does the conventional mutual information (Massey, 1990; Kramer, 1998).

We also draw connections with existing work on learning sub-task policies using imitation learning with the *options framework* (Sutton et al., 1998; Daniel et al., 2016). We show that our work, while derived using the information theoretic perspective of maximizing directed information, bears a close resemblance to applying the options framework in a generative adversarial imitation setting. Thus, our approach combines the benefits of learning hierarchical policies using the options framework with the robustness of generative adversarial imitation learning, helping overcome problems such as compounding errors that plague behaviour cloning.

In summary, the main contributions of our work include:

- We extend existing generative adversarial imitation learning frameworks to allow for learning of sub-task specific policies by maximizing directed information in a causal graph of sub-activity latent variables and observed trajectory variables.
- We draw connections between previous works on imitation learning with sub-task policies using *options* and show that our proposed approach can also be seen as *option* learning in a generative adversarial setting.
- We show through experiments on both discrete and continuous state-action spaces, the ability of our approach to segment expert demonstrations into meaningful sub-tasks and combine sub-task specific policies to perform the desired task.

## 2 RELATED WORK

### 2.1 IMITATION LEARNING

Imitation Learning (Pomerleau, 1989) aims at learning policies that can mimic expert behaviours from demonstrations. Modeling the problem as a Markov Decision Process (MDP), the goal in imitation learning is to learn a policy $\pi(a|s)$, which defines the conditional distribution over actions $a \in \mathcal{A}$ given the state $s \in \mathcal{S}$, from state-action trajectories $\tau = (s_0, a_0, \cdots, s_T)$ of expert behaviour. Recently,

Ho & Ermon (2016) introduced an imitation learning framework called Generative Adversarial Imitation Learning (GAIL) that is able to learn policies for complex high-dimensional physics-based control tasks. They reduce the imitation learning problem into an adversarial learning framework, for which they utilize Generative Adversarial Networks (GAN) (Goodfellow et al., 2014). The generator network of the GAN represents the agent's policy $\pi$ while the discriminator network serves as a local reward function and learns to differentiate between state-action pairs from the expert policy $\pi_{\mathbb{E}}$ and from the agent's policy $\pi$. Mathematically, it is equivalent to optimizing the following,

$$\min_{\pi} \max_{D} \mathbb{E}_{\pi}[\log D(s,a)] + \mathbb{E}_{\pi_E}[1 - \log D(s,a)] - \lambda H(\pi)$$

InfoGAIL (Li et al., 2017) and Hausman et al. (2017) solve the problem of learning from policies generated by a mixture of experts. They introduce a latent variable $c$ into the policy function $\pi(a|s,c)$ to separate different type of behaviours present in the demonstration. To incentivize the network to use the latent variable, they utilize an information-theoretic regularization enforcing that there should be high mutual information between $c$ and the state-action pairs in the generated trajectory, a concept that was first introduced in InfoGAN (Chen et al., 2016). They introduce a variational lower bound $L_1(\pi, Q)$ of the mutual information $I(c; \tau)$ to the loss function in GAIL.

$$L_1(\pi, Q) = \mathbb{E}_{c \sim p(c), a \sim \pi(\cdot|s,c)} \log Q(c|\tau) + H(c) \leq I(c; \tau)$$

The modified objective can then be given as,

$$\min_{\pi,q} \max_{D} \mathbb{E}_{\pi}[\log D(s,a)] + \mathbb{E}_{\pi_E}[1 - \log D(s,a)] - \lambda_1 L_1(\pi, q) - \lambda_2 H(\pi)$$

InfoGAIL models variations between different trajectories as the latent codes correspond to trajectories coming from different demonstrators. In contrast, we aim to model *intra-trajectory variations* and latent codes in our work correspond to sub-tasks (variations) within a demonstration. In Section 3, we discuss why using a mutual information based loss is infeasible in our problem setting and describe our proposed approach.

## 2.2 OPTIONS

Consider an MDP with states $s \in \mathcal{S}$ and actions $a \in \mathcal{A}$. Under the options framework (Sutton et al., 1998), an option, indexed by $o \in \mathcal{O}$ consists of a sub-policy $\pi(a|s,o)$, a termination policy $\pi(b|s,\bar{o})$ and an option activation policy $\pi(o|s)$. After an option is initiated, actions are generated by the sub-policy until the option is terminated and a new option is selected.

Options framework has been studied widely in RL literature. A challenging problem related to the options framework is to automatically infer options without supervision. Option discovery approaches often aim to find bottleneck states, *i.e.*, states that the agent has to pass through to reach the goal. Many different approaches such as multiple-instance learning (McGovern & Barto, 2001), graph based algorithms (Menache et al., 2002; Şimşek et al., 2005) have been used to find such bottleneck states. Once the bottleneck states are discovered, the above approaches find options policies to reach each such state. In contrast, we propose a unified framework using a information-theoretic approach to automatically discover relevant option policies without the need to discover bottleneck states.

Daniel et al. (2016) formulate the options framework as a probabilistic graphical model where options are treated as latent variables which are then learned from expert data. The option policies ($\pi(a|s,o)$) are analogous to sub-task policies in our work. These option policies are then learned by maximizing a lower bound using the Expectation-Maximization algorithm (Moon, 1996). We show how this lower bound is closely related to the objective derived in our work. We further show how this connection allows our method to be seen as a generative adversarial variant of their approach. Fox et al. (2017) propose to extend the EM based approach to multiple levels of option hierarchies. Further work on discovery of deep continuous options (Krishnan et al., 2017) allows the option policy to also select a continuous action in states where none of the options are applicable. Our proposed approach can also be extended to multi-level hierarchies (e.g. by learning VAEs introduced in section 3 with multiple sampling layers) or hybrid categorical-continuous macro-policies (e.g. using both categorical and continuous hidden units in the sampling layer in VAE).

Shiarlis et al. (2018) learn options by assuming knowledge of task sketches (Andreas et al., 2017) along with the demonstrations. The work proposes a behavior cloning based approach using connectionist temporal classification (Graves et al., 2006) to simultaneously maximize the joint likelihood of the sketch sequences and the sub-policies. Our proposed approach does not expect task sketches as input, making it more amenable to problems where labeling demonstrations with sketch labels is difficult.

Prior work in robot learning has also looked at learning motion primitives from unsegmented demonstrations. These primitives usually correspond to a particular skill and are analogous to options. Niekum & Barto (2011) used the Beta-Process Autoregressive Hidden Markov Model (BP-AR-HMM) to segment expert demonstrations and post-process these segments to learn motion primitives which provide the ability to use reinforcement learning for policy improvement. Alternately, Krishnan et al. (2018) use Dirichlet Process Gaussian Mixture Model (DP-GMM) to segment the expert demonstrations by finding transition states between linear dynamical segments. Similarly, Ranchod et al. (2015) use the BP-AR-HMM framework to initially segment the expert demonstrations and then use an inverse reinforcement learning step to infer the reward function for each segment. The use of appropriate priors allows these methods to discover options without a priori knowledge of the total number of skills. Kroemer et al. (2014) model the task of manipulation as an autoregressive Hidden Markov Model where the hidden phases of manipulation are learned from data using EM. However, unlike the above methods, in our proposed approach we also learn an appropriate policy over the extracted options. We show how this allows us to compose the individual option policies to induce novel behaviours which were not present in the expert demonstrations.

## 3 Proposed Approach

As mentioned in the previous section, while prior approaches can learn to disambiguate the multiple modalities in the demonstration of a sub-task and learn to imitate them, they cannot learn to imitate demonstrations of unsegmented long tasks that are formed by a combination of many small sub-tasks. To learn such sub-task policies from unsegmented deomonstrations we use the graphical model in Figure 1 (Right), *i.e.*, consider a set of expert demonstrations, each of which is represented by $\boldsymbol{\tau} = \{\tau_1, \cdots, \tau_T\}$ where $\tau_t$ is the state-action pair observed at time $t$. Each such demonstration has a corresponding sequence of latent variables $\boldsymbol{c} = \{c_1, \cdots, c_{T-1}\}$ which denote the sub-activity in the demonstration at any given time step.

As noted before, previous approaches (Li et al., 2017; Hausman et al., 2017) model the expert sub-task demonstrations using only a single latent variable. To enforce the model to use this latent variable, these approaches propose to maximize the mutual information between the demonstrated sequence of state-action pairs and the latent embedding of the nature of the sub-activity. This is achieved by adding a lower bound to the mutual information between the latent variables and expert demonstrations. This variational lower bound of the mutual information is then combined with the the adversarial loss for imitation learning proposed in Ho & Ermon (2016). Extending this to our setting, where we have a *sequence* of latent variables $\boldsymbol{c}$, yields the following lower bound on the mutual information,

$$L(\pi, q) = \sum_t \mathbb{E}_{c^{1:t} \sim p(c^{1:t}), a^{t-1} \sim \pi(\cdot|s^{t-1}, c^{1:t-1})} \Big[ \log q(c^t | c^{1:t-1}, \boldsymbol{\tau}) \Big] + H(\boldsymbol{c}) \leq I(\boldsymbol{\tau}; \boldsymbol{c}) \quad (1)$$

Observe that the dependence of $q$ on the entire trajectory $\boldsymbol{\tau}$ precludes the use of such a distribution at test time, where only the trajectory up to the current time is known. To overcome this limitation, in this work we propose to force the policy to generate trajectories that maximize the *directed* or *causal* information flow from trajectories to the sequence of latent sub-activity variables instead. As we show below, by using directed information instead of mutual information, we can replace the dependence on $\boldsymbol{\tau}$ with a dependence on the trajectory generated up to current time $t$.

The directed information flow from a sequence $\boldsymbol{X}$ to $\boldsymbol{Y}$ is given by,

$$I(\boldsymbol{X} \to \boldsymbol{Y}) = H(\boldsymbol{Y}) - H(\boldsymbol{Y} \| \boldsymbol{X})$$

where $H(\boldsymbol{Y} \| \boldsymbol{X})$ is the causally-conditioned entropy. Replacing $\boldsymbol{X}$ and $\boldsymbol{Y}$ with sequences $\boldsymbol{\tau}$ and $\boldsymbol{c}$,

$$
\begin{aligned}
I(\boldsymbol{\tau} \to \boldsymbol{c}) &= H(\boldsymbol{c}) - H(\boldsymbol{c} \| \boldsymbol{\tau}) \\
&= H(\boldsymbol{c}) - \sum_t H(c^t | c^{1:t-1}, \tau^{1:t}) \\
&= H(\boldsymbol{c}) + \sum_t \sum_{c^{1:t-1}, \tau^{1:t}} \Big[ p(c^{1:t-1}, \tau^{1:t}) \\
&\quad \sum_{c^t} p(c^t | c^{1:t-1}, \tau^{1:t}) \log p(c^t | c^{1:t-1}, \tau^{1:t}) \Big]
\end{aligned}
\tag{2}
$$

Here $\tau^{1:t} = (s_1, \cdots, a_{t-1}, s_t)$. A variational lower bound, $L_1(\pi, q)$ of the directed information, $I(\boldsymbol{\tau} \to \boldsymbol{c})$ which uses an approximate posterior $q(c^t | c^{1:t-1}, \tau^{1:t})$ instead of the true posterior $p(c^t | c^{1:t-1}, \tau^{1:t})$ can then be derived to get (See Appendix A.1 for the complete derivation),

$$
L_1(\pi, q) = \sum_t \mathbb{E}_{c^{1:t} \sim p(c^{1:t}), a^{t-1} \sim \pi(\cdot | s^{t-1}, c^{1:t-1})} \left[ \log q(c^t | c^{1:t-1}, \tau^{1:t}) \right] + H(\boldsymbol{c}) \le I(\boldsymbol{\tau} \to \boldsymbol{c})
\tag{3}
$$

Thus, by maximizing directed information instead of mutual information, we can learn a posterior distribution over the next latent factor $c$ given the latent factors discovered up to now and the trajectory followed up to now, thereby removing the dependence on the future trajectory. In practice, we do not consider the $H(\boldsymbol{c})$ term. This gives us the following objective,

$$
\min_{\pi, q} \max_D \; \mathbb{E}_\pi \left[ \log D(s, a) \right] + \mathbb{E}_{\pi_E} \left[ 1 - \log D(s, a) \right] - \lambda_1 L_1(\pi, q) - \lambda_2 H(\pi)
\tag{4}
$$

We call this approach Directed-Info GAIL. Notice that, to compute the loss in equation 3, we need to sample from the prior distribution $p(c^{1:t})$. In order to estimate this distribution, we first pre-train a variational auto-encoder (VAE) (Kingma & Welling, 2013) on the expert trajectories, the details of which are described in the next sub-section.

## 3.1 VAE PRE-TRAINING

Figure 2 (left) shows the design of the VAE pictorially. The VAE consists of two multi-layer perceptrons that serve as the encoder and the decoder. The encoder uses the current state $s_t$ and the previous latent variable $c_{t-1}$ to produce the current latent variable $c_t$. We used the Gumbel-softmax trick (Jang et al., 2016) to obtain samples of latent variables from a categorical distribution. The decoder then takes $s_t$ and $c_t$ as input and outputs the action $a_t$. We use the following objective, which maximizes the lower bound of the probability of the trajectories $p(\boldsymbol{\tau})$, to train our VAE,

$$
L_{\text{VAE}}(\pi, q; \boldsymbol{\tau_i}) = -\sum_t \mathbb{E}_{c^t \sim q} \left[ \log \pi(a^t | s^t, c^{1:t}) \right] + \sum_t D_{\text{KL}}(q(c^t | c^{1:t-1}, \tau^{1:t}) \| p(c^t | c^{1:t-1}))
\tag{5}
$$

Figure 2 (right) gives an overview of the complete method. The VAE pre-training step allows us to get approximate samples from the distribution $p(c^{1:t})$ to optimize equation 4. This is done by using $q$ to obtain samples of latent variable sequence $\boldsymbol{c}$ by using its output on the expert demonstrations. In practice, we fix the weights of the network $q$ to those obtained from the VAE pre-training step when optimizing the Directed-Info GAIL loss in equation 4.

## 3.2 CONNECTION WITH OPTIONS FRAMEWORK

In Daniel et al. (2016) the authors provide a probabilistic perspective of the options framework. Although, Daniel et al. (2016) consider separate termination and option latent variables ($b^t$ and $o^t$), for the purpose of comparison, we collapse them into a single latent variable $c^t$, similar to our framework with a distribution $p(c^t | s^t, c^{t-1})$. The lower-bound derived in Daniel et al. (2016) which is maximized using Expectation-Maximization (EM) algorithm can then be written as (suppressing dependence on parameters),

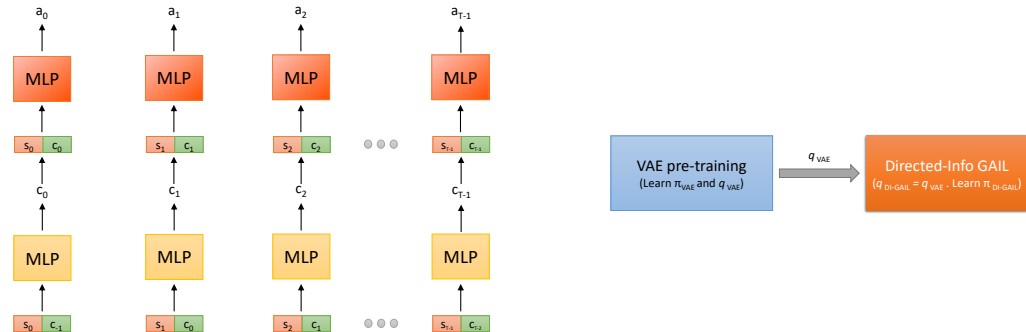

Figure 2: Left: VAE pre-training step. The VAE encoder uses the current state $(s_t)$, and previous latent variable $(c_{t-1})$ to produce the current latent variable $(c_t)$. The decoder reconstructs the action $(a_t)$ using $s_t$ and $c_t$. Right: An overview of the proposed approach. We use the VAE pre-training step to learn an approximate prior over the latent variables and use this to learn sub-task policies in the proposed Directed-Info GAIL step.

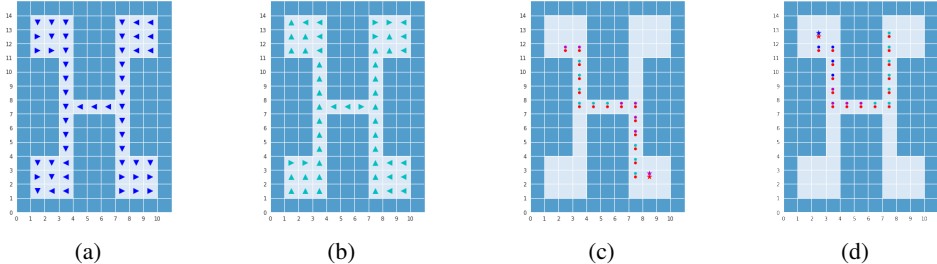

Figure 3: Results on the Four Rooms environment. (a) and (b) show results for two different latent variables. The arrows in each cell indicate the direction (action) with highest probability in that state and using the given latent variable. (c) and (d) show expert and generated trajectories in this environment. Star (*) represents the start state. The expert trajectory is shown in red. The color of the generated trajectory represents the latent code used by the policy at each time step.

$$p(\tau) \geq \sum_t \sum_{c^{t-1:t}} p(c^{t-1:t}|\tau) \log p(c^t|s^t, c^{t-1})) + \sum_t \sum_{c^t} p(c^t|\tau) \log \pi(a^t|s^t, c^t) \qquad (6)$$

Note that the first term in equation 6 *i.e.*, the expectation over the distribution $\log p(c^t|s^t, c^{t-1})$ is the same as equation 3 of our proposed approach with a one-step Markov assumption and a conditional expectation with given expert trajectories instead of an expectation with generated trajectories. The second term in equation 6 *i.e.*, the expectation over $\log \pi(a^t|s^t, c^t)$ is replaced by the GAIL loss in equation 4. Our proposed Directed-Info GAIL can be therefore be considered as the generative adversarial variant of imitation learning using the options framework. The VAE behaviour cloning pre-training step in equation 5 is exactly equivalent to equation 6, where we use approximate variational inference using VAEs instead of EM. Thus, our approach combines the benefits of both behavior cloning and generative adversarial imitation learning. Using GAIL enables learning of robust policies that do not suffer from the problem of compounding errors. At the same time, conditioning GAIL on latent codes learned from the behavior cloning step prevents the issue of mode collapse in GANs.

## 4 EXPERIMENTS

We present results on both discrete and continuous state-action environments. In both of these settings we show that (1) our method is able to segment out sub-tasks from given expert trajectories, (2) learn sub-task conditioned policies, and (3) learn to combine these sub-task policies in order to achieve the task objective.

| Environment | GAIL (Ho & Ermon, 2016) | VAE | Directed-Info GAIL |
|---|---|---|---|
| Pendulum-v0 | $-\mathbf{121.42} \pm \mathbf{94.13}$ | $-142.89 \pm 95.57$ | $-125.39 \pm 103.75$ |
| InvertedPendulum-v2 | $1000.0 \pm 15.23$ | $218.8 \pm 7.95$ | $\mathbf{1000.0} \pm \mathbf{14.97}$ |
| Hopper-v2 | $3623.4 \pm 51.0$ | $499.1 \pm 86.2$ | $\mathbf{3662.1} \pm \mathbf{21.7}$ |
| Walker2d-v2 | $4858.0 \pm 301.7$ | $1549.5 \pm 793.7$ | $\mathbf{5083.9} \pm \mathbf{356.3}$ |

Table 1: A comparison of returns for continuous environments. The returns were computed using 300 episodes. Our approach gives comparable returns to using GAIL but also segments expert demonstrations into sub-tasks. The proposed Directed-Info GAIL approach improves over the policy learned from the VAE pre-training step.

## 4.1 DISCRETE ENVIRONMENT

For the discrete setting, we choose a grid world environment which consists of a $15 \times 11$ grid with four rooms connected via corridors as shown in Figure 3. The agent spawns at a random location in the grid and its goal is to reach an apple, which spawns in one of the four rooms randomly, using the shortest possible path. Through this experiment we aim to see whether our proposed approach is able to infer sub-tasks which correspond to meaningful navigation strategies and combine them to plan paths to different goal states.

Figure 3 shows sub-task policies learned by our approach in this task. The two plots on the left correspond to two of the four different values of the latent variable. The arrow at every state in the grid shows the agent action (direction) with the highest probability in that state for that latent variable. In the discussion that follows, we label the rooms from 1 to 4 starting from the room at the top left and moving in the clockwise direction. We observe that the sub-tasks extracted by our approach represent semantically meaningful navigation plans. Also, each latent variable is utilized for a different sub-task. For instance, the agent uses the latent code in Figure 3(a), to perform the sub-task of moving from room 1 to room 3 and from room 2 to room 4 and the code in Figure 3(b) to move in the opposite direction. Further, our approach learns to successfully combine these navigation strategies to achieve the given objectives. For example, Figure 3(c, d) show examples of how the macro-policy switches between various latent codes to achieve the desired goals of reaching the apples in rooms 1 and 2 respectively.

## 4.2 CONTINUOUS ENVIRONMENTS

To validate our proposed approach on continuous control tasks we experiment with 5 continuous state-action environments. The first environment involves learning to draw circles on a 2D plane and is called *Circle-World*. In this experiment, the agent must learn to draw a circle in both clockwise and counter-clockwise direction. The agent always starts at (0,0), completes a circle in clockwise direction and then retraces its path in the counter-clockwise direction. The trajectories differ in the radii of the circles. The state $s \in \mathbb{R}^2$ is the (x,y) co-ordinate and the actions $a \in \mathbb{R}^2$ is a unit vector representing the direction of motion. Notice that in Circle-World, the expert trajectories include two different actions (for clockwise and anti-clockwise direction) for every state $(x, y)$ in the trajectory, thus making the problem multi-modal in nature. This requires the agent to appropriately disambiguate between the two different phases of the trajectory.

Further, to show the scalability of our approach to higher dimensional continuous control tasks we also show experiments on Pendulum, Inverted Pendulum, Hopper and Walker environments, provided in OpenAI Gym (Brockman et al., 2016). Each task is progressively more challenging, with a larger state and action space. Our aim with these experiments is to see whether our approach can identify certain action primitives which helps the agent to complete the given task successfully. To verify the effectiveness of our proposed approach we do a comparative analysis of our results with both GAIL (Ho & Ermon, 2016) and the supervised behavior cloning approaching using a VAE. To generate expert trajectories we train an agent using Proximal Policy Optimization (Schulman et al., 2017). We used 25 expert trajectories for the Pendulum and Inverted Pendulum tasks and 50 expert trajectories for experiments with the Hopper and Walker environments.

Figures 4(a, b, c) show results on the Circle-World environment. As can be seen in Figure 4(a, b), when using two sub-task latent variables, our method learns to segment the demonstrations into

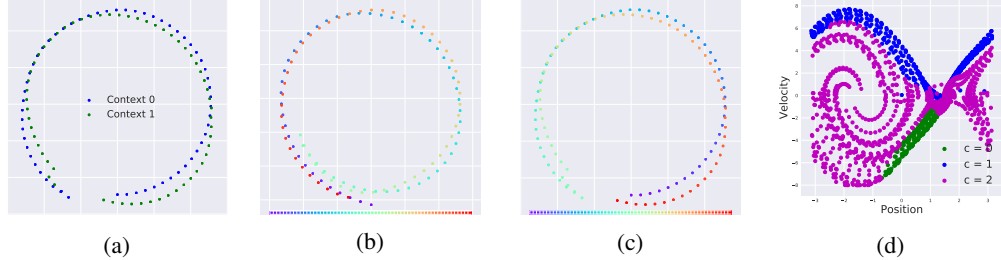

(a)             (b)             (c)             (d)

Figure 4: Results for Directed-Info GAIL on continuous environments. (a) Our method learns to break down the Circle-World task into two different sub-activities, shown in green and blue. (b) Trajectory generated using our approach. Color denotes time step. (c) Trajectory generated in opposite direction. Color denotes time step. (d) Sub-activity latent variables as inferred by Directed-Info GAIL on Pendulum-v0. Different colors represent different context.

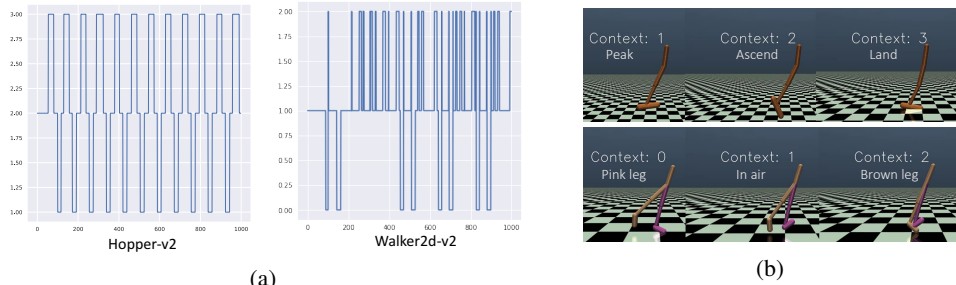

(a)                               (b)

Figure 5: (a) shows the plot of the sub-task latent variable vs time on the Hopper and Walker tasks. (b) shows discovered sub-tasks using Directed-Info GAIL on these environments.

two intuitive sub-tasks of drawing circles in clockwise and counterclockwise directions. Hence, our method is able to identify the underlying modes and thus find meaningful sub-task segmentations from unsegmented data. We also illustrate how the learned sub-task policies can be composed to perform *new* types of behavior that were unobserved in the expert data. In Figure 4(c) we show how the sub-task policies can be combined to draw the circles in inverted order of direction by swapping the learned macro-policy with a different desired policy. Thus, the sub-task policies can be utilized as a library of primitive actions which is a significant benefit over methods learning monolithic policies.

We now discuss the results on the classical Pendulum environment. Figure 4(d) shows the sub-task latent variables assigned by our approach to the various states. As can be seen in the figure, the network is able to associate different latent variables to different sub-tasks. For instance, states that have a high velocity are assigned a particular latent variable (shown in blue). Similarly, states that lie close to position 0 and have low velocity (i.e. the desired target position) get assigned another latent variable (shown in green). The remaining states get classified as a separate sub-task.

Figure 5 shows the results on the higher dimensional continuous control, Hopper and Walker, environments. Figure 5(a) shows a plots for sub-task latent variable assignment obtained on these environments. Our proposed method identifies basic action primitives which are then chained together to effectively perform the two locomotion tasks. Figure 5(b) shows that our approach learns to assign separate latent variable values for different action primitives such as, jumping, mid-air and landing phases of these tasks, with the latent variable changing approximately periodically as the agent performs the periodic hopping/walking motion.

Finally, in Table 1 we also show the quantitative evaluation on the above continuous control environments. We report the mean and standard deviations of the returns over 300 episodes. As can be seen, our approach improves the performance over the VAE pre-training step, overcoming the issue of compounding errors. The performance of our approach is comparable to the state-of-the-art GAIL (Ho & Ermon, 2016). Our method moreover, has the added advantage of segmenting the demonstrations into sub-tasks and also providing composable sub-task policies.

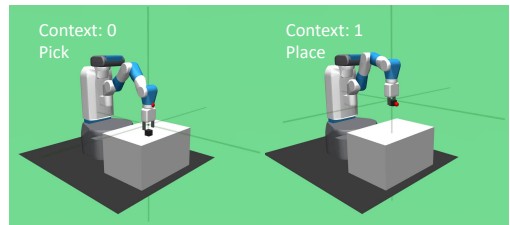

Figure 6: Segmentations obtained using our proposed Directed-Info GAIL method on FetchPickandPlace-v1.

| Method | Returns |
|---|---|
| VAE | $-14.07 \pm 5.57$ |
| GAIL | $-13.29 \pm 5.84$ |
| Directed-Info GAIL | $-11.74 \pm 5.87$ |
| GAIL + L2 loss | $-12.05 \pm 4.94$ |
| Directed-Info GAIL + L2 loss | $\mathbf{-9.47 \pm 4.84}$ |

Table 2: Mean returns over 100 episodes on FetchPickandPlace-v1 environment, calculated using the 'dense' reward setting.

We further analyze our proposed approach in more detail in the Appendix. In Appendix A.4 we visualize the sub-tasks in a low-dimensional sub-space. Also, in Appendix A.5 we show results when using a larger dimensional sub-task latent variable. A video of our results on Hopper and Walker environments can be seen at `https://sites.google.com/view/directedinfo-gail`.

### 4.3 OpenAI Robotics Environment

We further performed experiments on the FetchPickandPlace-v1 task in OpenAI Gym. In each episode of this task, the object and goal locations are selected randomly. The robot then must first reach and pick the object, and then move it to the goal location.

We trained agents using both our proposed Directed-Info GAIL and the baseline GAIL approaches. We used 500 expert demonstrations. While our method was able to learn to segment the expert demonstrations into the Pick and Place sub-tasks correctly, as can be seen in Figure 6 and the videos at `https://sites.google.com/view/directedinfo-gail/home#h.p_4dsbuC5expkZ`, neither our approach, nor GAIL was able to successfully complete the task. In our preliminary results, we found that the robot, in both our proposed approach and GAIL, would reach the object but fail to grasp it despite repeated attempts. To the best of our knowledge, no other work has successfully trained GAIL on this task either. Our preliminary experiments suggested that stronger supervision may be necessary to teach the agent the subtle action of grasping.

In order to provide this supervision, we additionally trained the policy to minimize the L2 distance between the policy action and the expert action on states in the expert demonstrations. At every training step, we compute the discriminator and policy (generator) gradient using the Directed-Info GAIL (or in the baseline, GAIL) loss using states and actions generated by the policy. Along with this gradient, we also sample a batch of states from the expert demonstrations and compute the policy gradient that minimizes the L2 loss between actions that the policy takes at these states and the actions taken by the expert. We weigh these two gradients to train the policy.

Table 2 shows the returns computed over 100 episodes. Adding the L2 measure as an additional loss led to significant improvement. Our proposed approach Directed-Info GAIL + L2 loss outperforms the baselines. Moreover, we believe that this quantitative improvement does not reflect the true performance gain obtained using our method. The reward function is such that a correct grasp but incorrect movement (e.g. motion in the opposite direction or dropping of the object) is penalized more than a failed grasp. Thus, the reward function does not capture the extent to which the task was completed.

Qualitatively, we observed a much more significant difference in performance between the proposed approach and the baseline. This can be seen in the sample videos of the success and failure cases for our and the baseline method at `https://sites.google.com/view/directedinfo-gail/home#h.p_qM39qD8xQhJQ`. Our proposed method succeeds much more often than the baseline method. The most common failure cases for our method include the agent picking up the object, but not reaching the goal state before the end of the episode, moving the object to an incorrect location or dropping the object while moving it to the goal. Agents trained using GAIL + L2 loss on the other hand often fail to grasp the object, either not closing the gripper or closing the gripper prematurely. We believe that our approach helps the agent alleviate this issue by

providing it with the sub-task code, helping it disambiguate between the very similar states the agent observes just before and just after grasping.

## 5 CONCLUSION

Learning separate sub-task policies can help improve the performance of imitation learning when the demonstrated task is complex and has a hierarchical structure. In this work, we present an algorithm that infers these latent sub-task policies directly from given unstructured and unlabelled expert demonstrations. We model the problem of imitation learning as a directed graph with sub-task latent variables and observed trajectory variables. We use the notion of directed information in a generative adversarial imitation learning framework to learn sub-task and macro policies. We further show theoretical connections with the options literature as used in hierarchical reinforcement and imitation learning. We evaluate our method on both discrete and continuous environments. Our experiments show that our method is able to segment the expert demonstrations into different sub-tasks, learn sub-task specific policies and also learn a macro-policy that can combines these sub-task.

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

## A  APPENDIX

### A.1  DERIVATION FOR DIRECTED-INFO LOSS

The directed information flow from a sequence $X$ to $Y$ is given by:

$$I(X \to Y) = H(Y) - H(Y \| X)$$

where $H(Y \| X)$ is the causally-conditioned entropy. Replacing $X$ and $Y$ with the sequences $\tau$ and $c$ give,

$$
\begin{aligned}
I(\tau \to c) &= H(c) - H(c \| \tau) \\
&= H(c) - \sum_t H(c^t | c^{1:t-1}, \tau^{1:t}) \\
&= H(c) + \sum_t \sum_{c^{1:t-1}, \tau^{1:t}} \left[ p(c^{1:t-1}, \tau^{1:t}) \sum_{c^t} p(c^t | c^{1:t-1}, \tau^{1:t}) \log p(c^t | c^{1:t-1}, \tau^{1:t}) \right] \\
&= H(c) + \sum_t \sum_{c^{1:t-1}, \tau^{1:t}} \left[ p(c^{1:t-1}, \tau^{1:t})[D_{KL}(p(\cdot|c^{1:t-1}, \tau^{1:t}) \| q(\cdot|c^{1:t-1}, \tau^{1:t})) \right. \\
&\qquad\qquad\qquad\qquad\qquad \left. + \sum_{c^t} p(c^t | c^{1:t-1}, \tau^{1:t}) \log q(c^t | c^{1:t-1}, \tau^{1:t})] \right] \\
&\geq H(c) + \sum_t \sum_{c^{1:t-1}, \tau^{1:t}} \left[ p(c^{1:t-1}, \tau^{1:t}) \sum_{c^t} p(c^t | c^{1:t-1}, \tau^{1:t}) \log q(c^t | c^{1:t-1}, \tau^{1:t}) \right].
\end{aligned}
$$

$$(7)$$

Here $\tau^{1:t} = (s_1, \cdots, a_{t-1}, s_t)$. The lower bound in equation 7 requires us to know the true posterior distribution to compute the expectation. To avoid sampling from $p(c^t|c^{1:t-1}, \tau^{1:t})$, we use the following,

$$
\begin{aligned}
\sum_{c^{1:t-1}} \sum_{\tau^{1:t}} &\left[ p(c^{1:t-1}, \tau^{1:t}) \sum_{c^t} p(c^t | c^{1:t-1}, \tau^{1:t}) \log q(c^t | c^{1:t-1}, \tau^{1:t}) \right] \\
&= \sum_{c^{1:t-1}} \sum_{\tau^{1:t}} \sum_{c^t} \left[ p(c^{1:t-1}, \tau^{1:t}) p(c^t | c^{1:t-1}, \tau^{1:t}) \log q(c^t | c^{1:t-1}, \tau^{1:t}) \right] \\
&= \sum_{c^{1:t-1}} \sum_{\tau^{1:t}} \sum_{c^t} \left[ p(c^t, c^{1:t-1}, \tau^{1:t}) \log q(c^t | c^{1:t-1}, \tau^{1:t}) \right] \\
&= \sum_{c^{1:t-1}} \sum_{\tau^{1:t}} \sum_{c^t} \left[ p(\tau^{1:t} | c^t, c^{1:t-1}) p(c^t, c^{1:t-1}) \log q(c^t | c^{1:t-1}, \tau^{1:t}) \right] \\
&= \sum_{c^{1:t}} p(c^{1:t}) \sum_{\tau^{1:t}} \left[ p(\tau^{1:t} | c^t, c^{1:t-1}) \log q(c^t | c^{1:t-1}, \tau^{1:t}) \right] \\
&= \sum_{c^{1:t}} p(c^{1:t}) \sum_{\tau^{1:t}} \left[ p(\tau^{1:t} | c^{1:t-1}) \log q(c^t | c^{1:t-1}, \tau^{1:t}) \right]
\end{aligned}
$$

$$(8)$$

where the last step follows from the causal restriction that future provided variables ($c^t$) do not influence earlier predicted variables ($\tau^{1:t}$ consists of states up to time $t$. $c_t$ does not effect state $s_t$). Putting the result in equation 8 in equation 7 gives,

$$L_1(\pi, q) = \sum_t \mathbb{E}_{c^{1:t} \sim p(c^{1:t}), a^{t-1} \sim \pi(\cdot|s^{t-1}, c^{1:t-1})} \left[ \log q(c^t | c^{1:t-1}, \tau^{1:t}) \right] + H(c) \leq I(\tau \to c) \quad (9)$$

|  | Directed Info-GAIL | | | VAE pre-training | |
| Environment | Epochs | Batch Size | posterior $\lambda$ | Epochs | Batch Size |
| --- | --- | --- | --- | --- | --- |
| Discrete | 1000 | 256 | 0.1 | 500 | 32 |
| Circle-World | 1000 | 512 | 0.01 | 1000 | 16 |
| Pendulum (both) | 2000 | 1024 | 0.01 | 1000 | 16 |
| Hopper-v2 | 5000 | 4096 | 0.01 | 2000 | 32 |
| Walker2d-v2 | 5000 | 8192 | 0.001 | 2000 | 32 |

Table 3: Experiment settings for all the different environments for both DirectedInfo-GAIL and VAE-pretraining step respectively.

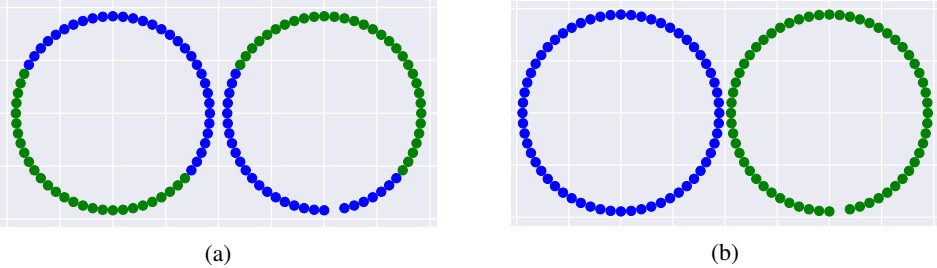

(a)                                                    (b)

Figure 7: Latent variable assignment on the expert trajectories in Circle-World (a) with and (b) without smoothing penalty $L_s$. Blue and green colors represent the two different values of the context variable. The centres of the two circles are shifted for clarity.

Thus, by maximizing directed information instead of mutual information, we can learn a posterior distribution over the next latent factor $c$ given the latent factors discovered up to now and the trajectory followed up to now, thereby removing the dependence on the future trajectory. In practice, we do not consider the $H(c)$ term. This gives us the objective,

$$\min_{\pi,q} \max_{D} \; \mathbb{E}_\pi[\log D(s,a)] + \mathbb{E}_{\pi_E}[1 - \log D(s,a)] - \lambda_1 L_1(\pi,q) - \lambda_2 H(\pi).$$

In practice, we fix $q$ from the VAE pre-training and only minimize over the policy $\pi$ in equation 4.

### A.2 Implementation Details

Table 3 lists the experiment settings for all of the different environments. We use multi-layer perceptrons for our policy (generator), value, reward (discriminator) and posterior function representations. Each network consisted of 2 hidden layers with 64 units in each layer and ReLU as our non-linearity function. We used Adam (Kingma & Ba, 2014) as our optimizer setting an initial learning rate of $3e^{-4}$. Further, we used the Proximal Policy Optimization algorithm (Schulman et al., 2017) to train our policy network with $\epsilon = 0.2$. For the VAE pre-training step we set the VAE learning rate also to $3e^{-4}$. For the Gumbel-Softmax distribution we set an initial temperature $\tau = 5.0$. The temperature is annealed using using an exponential decay with the following schedule $\tau = \max(0.1, \exp^{-kt})$, where $k = 3e - 3$ and $t$ is the current epoch.

### A.3 Circle-World smoothing

In the Circle-World experiment, we added another loss term $L_s$ to VAE pre-training loss $L_{VAE}$, which penalizes the number of times the latent variable switches from one value to another.

$$L_s = \sum_t \left[1 - \frac{c_{t-1} \cdot c_t}{\max(||c_{t-1}||_2, ||c_t||_2)}\right]$$

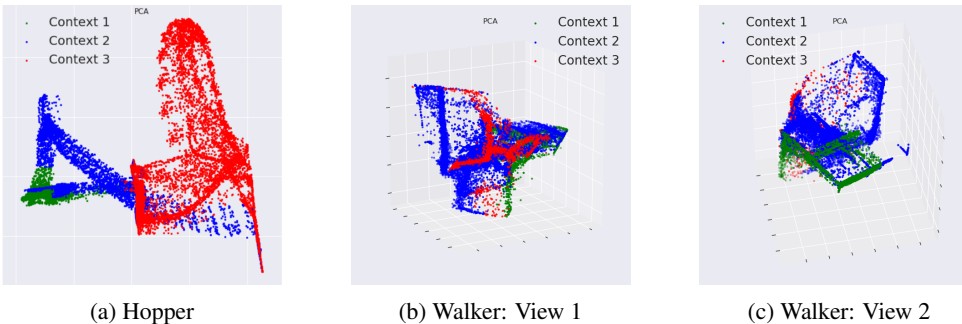

|  |  |  |
|:---:|:---:|:---:|
| (a) Hopper | (b) Walker: View 1 | (c) Walker: View 2 |

Figure 8: PCA Visualization for Hopper and Walker environment with sub-task latent variable of size 4.

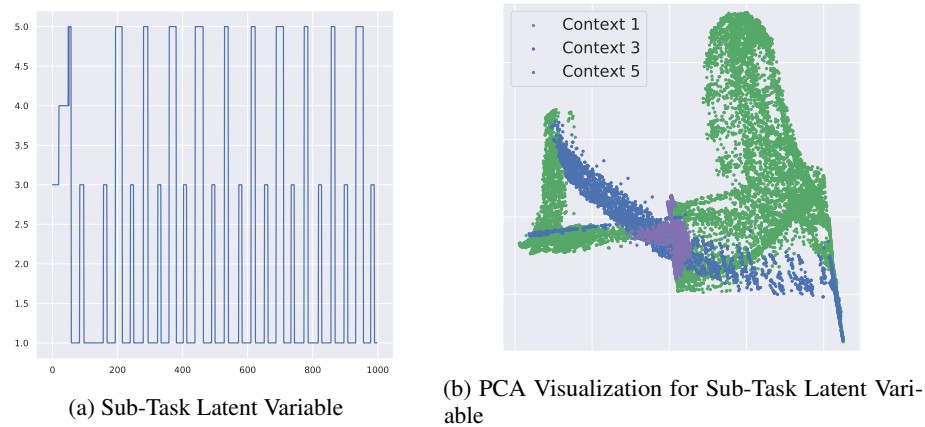

(a) Sub-Task Latent Variable  (b) PCA Visualization for Sub-Task Variable

Figure 9: Results on Hopper environment with sub-task latent variable of size 8.

Figure 7 shows the segmentation of expert trajectories with and without the $L_s$ term. We observed that without adding the smoothing penalty, the VAE learns to segment the expert trajectories into semi-circles as shown in Figure 7(a). While a valid solution, this does not match with the intuitive segmentation of the task into two sub-tasks of drawing circles in clockwise and counter-clockwise directions. The smoothing term can be thought of as a prior, forcing the network to change the latent variable as few times as possible. This helps reach a solution where the network switches between latent variables only when required. Figure 7(b) shows an example of segmentation obtained on expert trajectories after smoothing. Thus, adding more terms to the VAE pre-training loss can be a good way to introduce priors and bias solutions towards those that match with human notion of sub-tasks.

## A.4 PCA VISUALIZATION OF SUB-TASKS

In Figure 8, we show the plots expert states, reduced in dimensionality using Principal Component Analysis (PCA), in Hopper and Walker environments. States are color coded by the latent code assigned at these states. We reduced the dimension of states in Hopper from 11 to 2 and in Walker from 17 to 3. These low dimensional representations are able to cover $\sim 90\%$ of variance in the states. As can be seen in the figure, states in different parts of the space get assigned different latent variables. This further shows that our proposed approach is able to segment trajectories in such a way so that states that are similar to each other get assigned to the same segment (latent variable).

## A.5 USING LARGER CONTEXT

For the following discussion we will represent a $k$-dimensional categorical variable as belonging to $\Delta^{k-1}$ simplex. To observe how the dimensionality of the sub-task latent variable affects our proposed

approach we show results with larger dimensionality for the categorical latent variable $c_t$. Since DirectedInfo-GAIL infers the sub-tasks in an unsupervised manner, we expect our approach to output meaningful sub-tasks irrespective of the dimensionality of $c_t$. Figure 9 shows results for using a higher dimensional sub-task latent variable. Precisely, we assume $c_t$ to be a 8-dimensional one hot vector, i.e., $c_t \in \Delta^7$.

As seen in the above figure, even with a larger context our approach identifies similar basic action primitives as done previously when $c_t \in \Delta^3$. This shows that despite larger dimensionality our approach is able to reuse appropriate context inferred previously. We also visualize the context values for the low-dimensional state-space embedding obtained by PCA. Although not perfectly identical, these context values are similar to the visualizations observed previously for $c_t \in \Delta^3$. Thus our proposed approach is able, to some extent, infer appropriate sub-task representations independent of the dimensionality of the context variable.

