# OpenReview forum: "Directed-Info GAIL: Learning Hierarchical Policies from Unsegmented Demonstrations using Directed Information"
_ICLR.cc/2019/Conference_

### Official Review · AnonReviewer3 · 2018-10-19
**A well written paper with a relevant contribution for imitation learning**

**Rating:** 8
**Confidence:** 4

**Review:**

The paper describes a new learning framework, based on generative
adversarial imitation learning (GAIL), that is able to learn sub-tasks
policies from unsegmented demonstrations. In particular, it follows
the ideas presented in InfoGAIL, that depends on a latent variable,
and extend them to include a sequence of latent variables representing
the sequence of different subtasks. The proposed approach uses a
pre-training step, based on a variational auto-encoder (VAE), to
estimate latent variable sequences. The paper is well written and
relates the approach with the Options framework. It also shows,
experimentally, its performance against current state-of-the-art
algorithms.

Although the authors claim in the appendix that the approach is
relatively independent on the dimensionality of the context variable,
this statement needs further evidence. The approach is similar to HMMs
where the number f hidden states or latent variables can make a
difference in the performance of the system.

Also, it seems that the learned contexts do not necessarily correspond
to meaningful sub-tasks, as shown in the circle-world. In this sense,
it is not only relevant to determine the "right" size of the context
variable, but also how to ensure a meaningful sub-task segmentation.

---

> ### Author Response · Authors · 2018-11-14
> **Thank you for reviewing our paper**
>
> Thank you for your encouraging comments on the paper.
>
> We agree that further investigation on the dependence on the number of latent variables will be useful. Empirically, we found that the VAE often learns to ignore excess latent codes when the number of latent variables are close to the actual number of sub-tasks. For e.g. in the hopper and walker tasks even when the latent code size is set to 4, the VAE ends up only utilizing 3 codes. In the manipulation experiments we did on the suggestion of R2, when using 3 or 4 latent codes, the VAE only uses 2. These observations motivated us to perform the analysis that we report in the appendix. However, we agree that future work should analyse this further.
>
> The problem of discovering meaningful sub-tasks is certainly an interesting open problem. Using intuition driven loss functions, as we did in the circle world experiment, could be one way to allow networks to find sub-tasks meaningful to humans. Exploring other ways of introducing problem structure is definitely an important future direction.

---

### Official Review · AnonReviewer2 · 2018-11-06
**Review of Directed-Info GAIL**

**Rating:** 6
**Confidence:** 4

**Review:**

Summary:

This paper proposes an extension over the popular GAIL method for imitation learning  for the multi-modal data or tasks that have hierarchical structure in them. To achieve that the paper introduces an unsupervised variational objective by maximizing the directed mutual information between the latents c’s and the trajectories. The advantage of using directed information instead of regular MI based criterion is two-folds: 1) Being able to express the causal and temporal dependencies among the c’s changing across time. 2) Being able to learn a macro-policy without needing to condition on the future trajectories. Authors present results both on continuous and discrete environments.


Questions:
1) Can you give more detailed information about the hyperparameters of your model? For example how many seeds have you used?
2) Have you tried pre-training c_t’s as continuous latent variables?
3) Have you tried pre-training your model as Variational RNN instead of VAE?
4) Have you tried training your model on the pixels on the continuous control tasks?

Pros:
* Although the approach bears some similarity to Info-GAIL approach. The idea of using directed information for GAIL is novel and very interesting. This approach can be in particular useful for the tasks that have
* The paper is very well-written the goal and motivation of the paper is quite clear.

Cons:
* Experiments are quite weak. Both the discrete and the continuous environment experiments are conducted on very simplistic and toyish tasks. There are much more complicated and modern continuous control environments such as control suite [1] or manipulation suite [2].  In particular tasks where there is a more clear hierarchy would be interesting to investigate.
* Experimental results are underwhelming. For example Table 1, the results of the proposed approach is only barely better than the baseline.

[1] https://github.com/deepmind/dm_control
[2] Learning by Playing-Solving Sparse Reward Tasks from Scratch, M Riedmiller, R Hafner, T Lampe, M Neunert et al - arXiv preprint arXiv:1802.10567, 2018

---

> ### Author Response · Authors · 2018-11-14
> **Thank you for reviewing our paper**
>
> Thank you for your constructive comments on the paper.
>
> 1. Implementation details and hyper-parameter settings can be found in Table 2 and section A.2 of the appendix. The networks were trained for different number of iterations, and with different batch sizes for each environment as listed in the table. The networks were 2 layer MLPs, trained using Adam optimization with a learning rate of 3e-4. The VAEs were trained on the expert trajectories using the Gumbel-softmax trick with an exponential decay in temperature. The initial temperature was set to 5, and was decayed to around 0.1 by the end of the training. We used Proximal Policy Optimization for the policy updates while optimizing the Directed-Info Loss. Batch sizes for all environments are listed in table 2. In all experiments, the number of expert episodes were selected to approximately have an equal number of generated and expert state-action pairs in a batch. The lambda parameter was set differently for each environment and these settings can be seen in table 2 (posterior lambda column). We used 4 latent codes in the room environment, 2 in circle world, and 3 in each of the Mujoco environments. We used 5 different seeds in the Open AI gym environments. The results were computed by averaging over 300 episodes.
>
> 2. While we did not try pre-training c_{t}s as continuous variables, we use temperature annealing, starting with a high initial temperature of 5, which is decreased over the epochs of VAE training to around 0.1. This means that during the initial epochs of the training, the latent variables are continuous, and only later in the training do they approximate categorical variables. Also, as noted in response to R1, we also tried using continuous latent variables in early experiments. Although the this led to a lower L2 loss during VAE training, the high representational power of continuous variables meant that the network learned to assign different latent codes to sub-tasks which were intuitively similar but in different states. Since our goal was to learn sub-task specific policies, we switched to using discrete latent variables. By forcing the network to use only 1-of-n possible codes, the network was forced to assign the same code to similar behaviors, even if they occur in different states.
>
> 3. We did try using variational RNNs during early experiments on simple discrete environments (that we did not report here). We did not find much advantage in using recurrent models over providing state history to an MLP in those environments, and hence all later experiments were done using feed-forward architectures with history of appropriate time length.
>
> 4. No, we haven't tried training our models on pixels for the continuous control tasks. However, in principle, this can be done using convolutional variational autoencoders during the VAE step and then using CNNs for generator and discriminators (similar to Deep Convolutional GAN).
>
> Experiments on OpenAI Robotics environments - Following suggestions, we tried to test the baselines and our approach on the FetchPickandPlace task in OpenAI Gym. While our method was able to learn to segment the expert demonstrations into the Pick and Place sub-tasks correctly, as can be seen in the videos at https://sites.google.com/view/directedinfo-gail/home#h.p_4dsbuC5expkZ , neither our approach, nor GAIL was able to successfully complete the task. In our preliminary results, we found that the robot, in both our proposed approach and GAIL, would reach the object but fail to grasp it despite repeated attempts. To the best of our knowledge, no other work has successfully trained GAIL on this task either. Our preliminary experiments seem to suggest that stronger supervision may be necessary to teach the agent the subtle action of grasping.
>
> Experiments on problems with hierarchical structure - We had also tried some experiments on tasks with clearer hierarchical structure prior to submission. We constructed rewards for a monoped agent to perform three different subtasks - walk forward, walk backward and jump. Then, we trained RL agents to perform 2 of these sub-tasks one after the other in an episode. We found that RL agents with MLP policies trained using PPO and DDPG failed to learn any combination of these sub-tasks. We were able to train agents using phase-functioned policies [1, 2] to perform 4 of 6 combinations. However, we found that the gait of the agent was strongly dependent on the ordering of the sub-tasks. This made identifying common sub-tasks hard. We found training phase policies for imitation learning with such noisy segmentations to be challenging. We believe that this is beyond the scope of the paper and should be left to future work.
>
> [1] Phase-Functioned Neural Networks for Character Control. ACM Transactions on Graphics, 2017
> [2] Phase-Parametric Policies for Reinforcement Learning in Cyclic Environments. AAAI 2018

---

> > ### Author Response · Authors · 2018-11-24
> > **Updated results on OpenAI Robotics environment**
> >
> > As mentioned in our previous comment, we observed that agents trained with both GAIL and our proposed approach failed to learn to grasp the object, pointing to a need for stronger supervision, especially at states close to the grasping states. We performed some more experiments, where we provide this supervision by training the policy to minimize the L2 distance between the policy action and the expert action on states in the expert demonstrations.
> >
> > At every training step, we compute the discriminator and policy (generator) gradient using the Directed-Info GAIL (or in the baseline, GAIL) loss using states and actions generated by the policy. Along with this gradient, we also sample a batch of states from the expert demonstrations and compute the policy gradient that minimizes the L2 loss between actions that the policy takes at these states and the actions taken by the expert. We weigh these two gradients to train the policy.
> >
> > The mean and standard deviation of returns on 100 episodes is as follows (higher is better)-
> >
> > Directed Info GAIL + L2 loss: Mean = -9.47, Std dev. = 4.84
> > GAIL + L2 loss: Mean = -12. 05, Std dev. = 4.94
> > Directed-Info GAIL: Mean = -11.74, Std dev. = 5.87
> > GAIL: Mean = -13.29, Std dev. = 5.84
> >
> > Adding the L2 measure as an additional loss led to significant improvement. Our proposed approach Directed-Info GAIL + L2 loss outperforms the baseline. Moreover, we believe that this quantitative improvement does not reflect the true performance gain obtained using our method. The reward function is such that a correct grasp but incorrect movement (e.g. motion in the opposite direction or dropping of the object) is penalized more than a failed grasp. Thus, the reward function does not capture the extent to which the task was completed.
> >
> > Qualitatively, we observed a much more significant difference in performance between the proposed approach and the baseline. This can be seen in the sample videos of the success and failure cases for our and the baseline method at https://sites.google.com/view/directedinfo-gail/home#h.p_qM39qD8xQhJQ
> >
> > We observed that our proposed method succeeds much more often than the baseline method. The most common failure cases for our method include the agent picking up the object, but not reaching the goal state before the end of the episode, moving the object to an incorrect location or dropping the object while moving it to the goal. Agents trained using GAIL + L2 loss on the other hand often fail to grasp the object, either not closing the gripper or closing the gripper prematurely. We believe that our approach helps the agent alleviate this issue by providing it with the sub-task code, helping it disambiguate between the very similar states the agent observes just before and just after grasping.

---

> > > ### Author Response · Authors · 2018-11-26
> > > **Updated results on OpenAI Robotics environment**
> > >
> > > For completeness, here is the table of results on the FetchPickandPlace-v1 environment with results of the VAE baseline included:
> > >
> > > Directed Info GAIL + L2 loss: Mean = -9.47, Std dev. = 4.84
> > > GAIL + L2 loss: Mean = -12. 05, Std dev. = 4.94
> > > Directed-Info GAIL: Mean = -11.74, Std dev. = 5.87
> > > GAIL: Mean = -13.29, Std dev. = 5.84
> > > VAE: Mean = -14.07, Std dev. = 5.57

---

### Official Review · AnonReviewer1 · 2018-11-06
**The InfoGail method extended to online latent code estimation at test time**

**Rating:** 6
**Confidence:** 4

**Review:**

The paper presents a learning-based method for learning the latent context codes from demonstrations along with a GAIL model.
This amounts to learning the option segments and the policies simultaneously.
The main contribution is the model the problem as a time-dependent context and then use a directed information flow loss instead of the mutual information loss.

1. What is the effect of models of the underlying distribution of latent codes.
Can it be categorical only, or can it be continuous?
Could we also model it as multidimensional?
The current results only provide single dimensional categorial distribution as latent codes.

2. The paper missed an important line of work which solves nearly the same problem -- option discovery and policy learning.
Krishnan -- Discovery of Deep Option(1703.08294). This work was used by authors in continuous options and then again for program generation (https://openreview.net/pdf?id=rJl63fZRb).

They explicitly infer the option parameters, along with termination conditions with the Expectation Propagation method.
The results are in very similar domains hence comments, if not a comparison, would be useful.


3. The authors state that the main problem with an InfoGail style method is dependence on the full trajectory as in eq 1. Hence the directed info flow is required to solve the problem. However in the actual model, the authors make a sequence of variational approximations -- (a) reduction of eq2 to eq1 with a variation lower bound on posterior p(c|c,\tau) and then replace the prior p(c) with q(c|c,\tau) in eq 5. But looking at the model diagram in fig 2. the VAE actually makes the Markovian assumption -- i.e. c only depends on c_{t-1} and s_{t}. If that is true then how would this be very different from InfoGAIL mutual information loss.
It appears that to capture the authors' mathematical intuition the VAE should have a recurrent generator which should have a hidden state factor passing in to capture dependence on history until the current time.

3a. In fact the first term in eq 6 looks closer to the actually used model. If that is not true then the authors should clarify.

4. Experiments do capture the notion discovery of options. But the simplicity of data leaves much to be desired.
One of the main difference of this work in comparison to unsupervised segmentation models GMM or BP-AR-HMM is the fact that the options learned are composable. But the authors only show this composability on the circle domain -- which is arguably a toy-domain.
A reasonable confirmation that the model indeed learns composition is to generate a trajectory for a sequence of latent code not seen in data. -- like walking -- normal -- left-right-left can be converted to limping gait -- left-left-right-right. This is only a suggestive example.

5. In appendix eq 8 how is the reduction from line 3 to line 4 of the equation made -- what is the implicit assumption.
joint distribution p(c, \tau) is written out as p (\tau|c) p(c) without an integral.

---

> ### Author Response · Authors · 2018-11-14
> **Thank you for reviewing our paper**
>
> Thank you for your feedback on our paper.
>
> 1. The latent codes can be categorical or continuous, single or multi-dimensional. Since our approach utilizes a VAE, any distribution which allows for sampling using the reparameterization trick can be used. Our approach does not put any additional constraints. While we reported results with categorical latent codes, we also experimented with continuous variables with multi-dimensional Gaussian priors in early experiments. Using continuous latent variables often allowed faster training and lower loss for the VAE step due to their rich representational power. However, using continuous variables as options has a drawback because the network can select very different values of context for seemingly same sub-tasks in different states. This goes against the idea of having sub-task specific policies, as now the latent codes become much more susceptible to the state as opposed to the sub-task. By discretizing the context into 1-of-n values using categorical variables, we found that the sub-tasks correlate much better with sub-tasks that are intuitive to humans.
>
> 2. Thank you for pointing us to the line of work on Discovery of Deep Options (DDO). While very relevant, their approach is different from our proposed approach and is similar to the work from Daniel et al. we cited. DDO proposes to extend the EM based approach to multiple levels of option hierarchies. Their work on Discovery of Deep Continuous Options allows the option policy to also select a continuous action in states where none of the options are applicable. Here, we would like to point out that the title of the paper is somewhat misleading since the options are still modeled as categorical variables in their paper and are not continuous. Note that their approaches belong to the domain of behavior cloning. In contrast, we propose a method to integrate GAIL with the options framework. GAIL and other works that build on it, use Inverse Reinforcement Learning to learn policies as well as rewards, overcoming problems such as compounding errors, and have been shown to need fewer expert demonstrations than behavior cloning. Moreover, our proposed approach can also be extended to multi-level hierarchies (e.g. by learning VAEs with multiple sampling layers) or hybrid categorical-continuous macro-policies (e.g. using both categorical and continuous hidden units in the sampling layer in VAE). We will add this discussion to the paper.
>
> 3. We would like to clarify that Eq 1 is not the loss used in Info-GAIL. The graphical model used in Info-GAIL (Fig. 1 left) does not model sub-tasks, while in contrast Fig. 1 (right) shows the graphical model that we propose to use. This enables us to model expert demonstrations as an interaction between sub-tasks and their resulting state-action trajectory, and learn re-usable sub-task specific policies. Eq 1 is the loss function under this graphical model when using mutual information. The equation can be modified to have a Markov assumption where c_{t} only depends on c_{t-1}. However, the dependence on future states still remains since that is by ‘definition’ of mutual information, which cannot be altered using a Markov assumption. Markov assumption only allows us to remove dependence on the past (not future) given the most recent history. The assumption you propose is precisely the effect of utilizing directed information. We will make this point clearer.
>
> We agree that a recurrent model better captures the mathematical intuition. In the room and circle world tasks, the encoder (but not the decoder/policy) did get a history of previous 5 time steps as an approximation for the trajectory until the current time step. In the Mujoco environments, we found that the state was representative enough to not make a difference in practice.
>
> 4. While we agree that generating unseen gaits would be ideal, unfortunately, the one time-step option paradigm makes unseen composition hard. This difficulty is further compounded by demonstrations with unnatural and asymmetric gaits (e.g. the two feet of the agent do not support the agent’s motion equally and play different roles). We would also argue that it is not easy to decompose a walk into 2 separate limping motions, just as two independent limping movements do not fully constitute a walk. However, we provide more evidence that the policy does indeed use the different latent variables to perform different sub-tasks. We have uploaded two new videos at https://sites.google.com/view/directedinfo-gail/home#h.p_cEMQy28s4Jkb where we give just one latent code to the policy in each video - code to put pink leg down in video 1, and code to use the brown leg in video 2. As can be seen, both latent codes give rise to different behaviors. Please also see response to R2 on experiments on environments with clearer hierarchies.
>
> 5. This is by the rule P(A, B) = P(A|B)P(B). We will make the steps in the derivation clearer.

---

### Author Response · Authors · 2018-11-25
**General reply to reviewers and area chair**

We would like to thank the reviewers for their constructive feedback on our paper. We are encouraged by the positive reviews. The reviewers noted that our work makes a relevant contribution and that our approach is novel and interesting. They agreed unanimously that the paper is clearly written and well motivated. R1 and R2 recommended performing experiments on more complicated benchmark tasks. Given this feedback, we performed further experiments on a challenging recent manipulation environment from OpenAI gym. We discuss the results of these experiments in detail in a comment to R2 and have also added these details to the appendix section of the paper. Our results clearly demonstrate the merits of our approach over state-of-the-art baselines. We will integrate these new results and include the suggested improvements to the literature review in the final version of the paper.

We hope that the reviewers and the area chair will take these new experiments into account when assessing the final scores.

---

### Meta-Review · Area_Chair1 · 2018-12-14

**Confidence:** 4
**Recommendation:** Accept (Poster)

**Metareview:**

This paper proposes an approach for imitation learning from unsegmented demonstrations. The paper addresses an important problem and is well-motivated. Many of the concerns about the experiments have been addressed with follow-up comments. We strongly encourage the authors to integrate the new results and additional literature to the final version. With these changes, the reviewers agree that the paper exceeds the bar for acceptance. Thus, I recommend acceptance.